# Assessing the Causal Relationship Between Plasma Proteins and Pulmonary Fibrosis: A Systematic Analysis Based on Mendelian Randomization

**DOI:** 10.3390/biology14020200

**Published:** 2025-02-14

**Authors:** Moxuan Han, Yan Cui, Zhengyuan Fang, He Li, Yueqi Wang, Mingwei Sima, Yan Bi, Donghui Yue

**Affiliations:** 1School of Traditional Chinese Medicine, Changchun University of Chinese Medicine, Changchun 130117, China; 23102570153@stu.ccucm.edu.cn (M.H.); lihetcm@163.com (H.L.); haibiandeyilisha@163.com (Y.W.); smmwyjs@163.com (M.S.); biyan407@126.com (Y.B.); 2School of Basic Medicine, Changchun University of Chinese Medicine, Changchun 130117, China; cycczyydx@163.com (Y.C.); fangzhengyuan94@163.com (Z.F.)

**Keywords:** drug targets, Mendelian randomization, pulmonary fibrosis, plasma proteins

## Abstract

Pulmonary fibrosis (PF) is a serious lung disease that leads to breathing problems and, if left untreated, can cause respiratory failure. Scientists know that certain proteins in the blood may be linked to PF, but there hasn’t been much research into how these proteins influence the disease. This study aimed to identify key proteins that could be potential targets for new PF treatments. Using a method called Mendelian randomization (MR), the researchers found 64 proteins that are strongly connected to PF. The study also looked at how these proteins interact with each other and identified several drugs that could potentially be used to treat PF, such as sorafenib, vitamin C, and vitamin E. Additionally, the researchers confirmed that these proteins are highly active in certain lung cells, supporting their role in PF. The findings open up new opportunities for creating targeted treatments for PF, which could lead to faster, more effective therapies for patients and lower development costs for drug companies.

## 1. Introduction

Pulmonary fibrosis (PF) is a complex interstitial lung disease marked by the destruction of alveolar structures, abnormal extracellular matrix (ECM) deposition, and the progressive stiffening of lung tissue, which can eventually result in respiratory failure [1]. PF is a chronic and progressive disease with early symptoms that are often subtle. However, its high mortality and low survival rates have made it a critical focus of clinical attention [2]. Epidemiological studies show that the incidence of PF is rising globally, severely impacting patients’ quality of life and placing a significant burden on healthcare resource [3]. The global aging population is accelerating, leading to an annual increase in PF cases, which has become an urgent public health concern [4]. PF includes diverse pathological subtypes, showing marked heterogeneity in clinical outcomes and pathogenic mechanisms. For instance, idiopathic pulmonary fibrosis (IPF) is characterized histopathologically by the usual interstitial pneumonia (UIP) pattern [5]. Its distinctive honeycombing and rapidly progressing fibrosis lead to a particularly poor prognosis. In contrast, non-idiopathic forms, such as non-specific interstitial pneumonia (NSIP), have a slower disease progression and more favorable outcomes [6]. Molecular studies reveal key differences between UIP and NSIP in gene expression profiles, inflammatory mediator regulation, and ECM remodeling, suggesting their involvement in distinct pathogenic networks [7]. Despite significant progress in understanding the pathogenesis of PF, current therapeutic approaches remain limited and ineffective in halting disease progression. Therefore, exploring the underlying mechanisms of PF and identifying effective therapeutic targets are essential for developing new treatment strategies.

Proteins are essential regulatory factors in all biological processes and are closely linked to the onset and progression of diseases [8]. The human plasma proteome consists of proteins secreted into the circulatory system or entering the bloodstream, which perform functions in circulation and are crucial for inter-tissue communication [9]. Under pathological conditions, the expression and function of the plasma proteome are often disrupted, making it a potential source of disease biomarkers and therapeutic targets [10]. Due to the easier accessibility of blood compared to other tissues, the plasma proteome is a valuable resource for identifying molecular markers of diseases in large-scale populations. Existing studies show that plasma proteins play significant roles in PF, with some proteins potentially involved in its pathogenesis and progression [11]. However, current evidence is still limited and lacks comprehensive and systematic causal studies. Moreover, the inherent confounding bias and reverse causation in observational studies reduce the reliability of research conclusions. Although certain proteins may be identified as biomarkers, they are not necessarily direct causes of the disease. Therefore, more robust causal inference methods are needed to explore the specific roles of the plasma proteome in PF.

Advances in large-scale proteomics now enable the simultaneous analysis of thousands of plasma proteins in large studies [12]. Integrating plasma proteomics with genome-wide association studies (GWASs) allows researchers to identify protein quantitative trait loci (pQTLs) [13]. Pairing these pQTLs with disease-associated genetic variants enables Mendelian randomization (MR) to determine the causal relationships between proteins and diseases [13]. MR, an established tool in genetic epidemiology, uses genetic variants as instrumental variables to reduce confounding and reverse causality, enhancing the reliability of causal inferences [14]. Using pQTLs as instrumental variables enables MR to perform the high-throughput screening of plasma proteins and identify those causally linked to PF. Furthermore, assessing the druggability of these proteins and their clinical potential is essential.

This study aims to systematically identify plasma proteins and potential drug targets causally associated with PF through an integrated MR framework. By combining drug prediction and molecular docking analyses, the pharmacological activity of these targets is validated, thus expanding the clinical applicability of candidate drugs. Additionally, the study integrates single-cell sequencing technology to analyze the expression patterns and functions of key proteins across different lung cell subpopulations [15]. Enrichment analysis and protein–protein interaction (PPI) networks are also used to uncover the biological significance of these targets and their roles in key PF pathological processes. Through this multi-faceted approach, this study not only deepens the understanding of PF pathogenesis but also provides a theoretical foundation and practical guidance for developing core protein-based precision therapies, advancing PF prevention and personalized medicine.

## 2. Methods

This study uses an integrated approach to assess the causal relationship between plasma proteins and PF. Additionally, the study explores the biological significance of plasma proteins and their potential for drug development. Figure 1 presents the overall framework of this research.

### 2.1. Data Sources

#### 2.1.1. Exposure Data

Genetic summary statistics associated with plasma protein levels were obtained from pQTL studies by Ferkingstad et al., involving 35,559 individuals of Icelandic descent (deCODE Database, https://www.decode.com/summarydata/ [accessed on 6 November 2024]) [13].

#### 2.1.2. Outcome Data

The dataset for this study was sourced from the publicly available GWAS study (IEU GWAS Database, https://gwas.mrcieu.ac.uk/ [accessed on 6 November 2024]) [16] (ebi-a-GCST90018908), which focuses on genetic data associated with PF. This dataset includes 1566 cases and 467,560 controls, covering 24,195,349 SNPs, mainly from European populations [17]. However, due to the lack of subtype classification for PF in the dataset, subtype-specific stratified analyses could not be performed in this study. Despite significant differences in molecular mechanisms and distinct clinical prognoses between UIP and NSIP, this study aimed to identify molecular mechanisms related to PF through systematic analysis, without further stratifying these subtypes.

#### 2.1.3. Mendelian Randomization Analysis

This study utilized a bidirectional MR framework to assess the causal link between plasma proteins and PF [18]. Exposure data were obtained from preprocessed pQTL summary statistics. During instrument selection, the explained variance and F-statistic (F > 10) of each SNP were calculated. SNPs with an effect allele frequency (EAF) > 0.01 were retained, while linkage disequilibrium (LD) clustering was disabled to reduce redundant signals. Outcome data were retrieved from the EBI database (ID: ebi-a-GCST90018908) and deduplicated. A standardized data harmonization process was applied to align exposure and outcome variables, involving chromosome position matching, allele orientation adjustment, and the exclusion of palindromic SNPs. Furthermore, SNPs directly associated with PF (*p*-value for the outcome > 0.05) were excluded to mitigate confounding bias. Causal effects were estimated using the inverse-variance weighted (IVW) method, MR-Egger regression, weighted median method, and simple/weighted mode method, with odds ratios (ORs) and their corresponding 95% confidence intervals calculated. Sensitivity analyses comprised the Cochran’s Q test (for heterogeneity assessment), MR-Egger intercept test (for horizontal pleiotropy detection), leave-one-out analysis (for robustness evaluation), and funnel plots (for symmetry assessment). For IVW results demonstrating statistical significance (*p* < 0.05), four diagnostic plots were generated (scatter plot, forest plot, funnel plot, and leave-one-out plot) (Appendix A). The final selection criteria required consistent OR directions across all five methods (either all >1 or all <1), the exclusion of significant MR-Egger intercept results (*p* < 0.05), and the retention of the most significant measurements in repeated protein assessments.

In the reverse analysis, SNPs linked to PF were identified using the genome-wide significance threshold (*p* < 5 × 10^−8^). LD clustering (r^2^ < 0.001) was employed to minimize correlations among instruments, and the instrument strength (F > 10) was revalidated to ensure the robustness of bidirectional causality. All analyses were performed in the R 4.4.1 environment using the TwoSampleMR package, with visualizations and statistical details available in Appendix A.

### 2.2. Enrichment Analysis

This study investigated the biological functions of potential therapeutic targets using Gene Ontology (GO) and Kyoto Encyclopedia of Genes and Genomes (KEGG) enrichment analyses. GO enrichment analyses (covering Biological Processes [BPs], Molecular Functions [MFs], and Cellular Components [CCs]) and KEGG pathway enrichment were performed on significant genes identified via Mendelian randomization, using the R package ClusterProfiler [19]. Terms with a raw *p*-value < 0.05 were retained and adjusted using the Benjamini–Hochberg correction method (refer to Appendix A). The results were presented as bubble plots for visualization.

### 2.3. Construction of Protein–Protein Interaction Network and Screening of Core Genes

To better understand the interactions between proteins within the cell, a protein–protein interaction (PPI) network was constructed using the STRING database (https://cn.string-db.org/ [accessed on 11 November 2024]). A confidence score of 0.15 was set as the minimum required interaction score, and disconnected nodes were excluded [20]. All other parameters were kept as default. The resulting file was imported into Cytoscape V.3.8.0 for visualization. The cytoHubba plugin in Cytoscape was used to filter the top 10 genes based on their degree values as core genes [21]. Nodes were color-coded according to their degree values, with redder colors indicating higher degree values.

### 2.4. Drug Prediction

Evaluating the interaction between proteins and drugs is a critical step in determining whether a target protein can serve as a viable drug target. In this study, we explored the relationship between core genes and potential drugs using the Drug Signatures Database (DSigDB, http://dsigdb.tanlab.org/DSigDBv1.0/ [accessed on 15 November 2024]) [22]. The DSigDB database includes over 22,500 gene sets, more than 17,000 compounds, and nearly 20,000 genes, facilitating effective association between drugs and their target genes. It is a valuable resource for drug target research and drug discovery. Specifically, we uploaded the selected core protein gene information to DSigDB to predict candidate drugs that bind to the target genes, providing a theoretical foundation for targeted gene therapy. Building on this, we conducted drug enrichment analysis using the R package ClusterProfiler. During the analysis, we set significance criteria with *p*-values and adjusted *p*-values (p.adjust) both set at 0.05 to select drugs significantly associated with core genes. The enrichment analysis employed the hypergeometric test method to evaluate whether core genes were significantly enriched in the target gene sets of specific drugs, ultimately selecting significant results with *p*-values less than 0.05. Bar charts and gene–drug relationship network diagrams were generated to display the ranking of enriched drugs, gene proportions, and the relationships between core genes and drugs, respectively.

### 2.5. Molecular Docking

To better understand the impact of candidate drugs on drug target proteins and their drug availability, we performed molecular docking to evaluate the binding affinity and interaction patterns between the candidate drugs and their targets. Based on the results of the previous drug enrichment analysis, the top five candidates were selected for molecular docking analysis. Molecular docking was performed using the CB-Dock2 platform (https://cadd.labshare.cn/cb-dock2/index.php [accessed on 8 December 2024]) [23]. Drug structure data were obtained from the PubChem compound database (https://pubchem.ncbi.nlm.nih.gov/ [accessed on 8 December 2024]). Protein structure data were obtained from the Protein Data Bank (PDB, https://www.rcsb.org/ [accessed on 8 December 2024]).

### 2.6. Single-Cell Sequencing

To further investigate the expression characteristics and potential functions of the selected core genes in PF, we conducted single-cell sequencing analysis. We obtained a single-cell RNA sequencing dataset of human lung proximal airway mesenchymal cells (SRA640325:SRS2769051) from the PanglaoDB platform (https://panglaodb.se/ [accessed on 10 December 2024]) [24]. This dataset provides gene expression data for human lung proximal airway mesenchymal cells at the single-cell level. We integrated the single-cell data to analyze the expression of these core genes in lung proximal airway mesenchymal cells, aiming to identify specific cell populations that may play a role in the development of PF. We used the t-SNE dimensionality reduction method to visualize the single-cell data and observe the distribution patterns of core genes across various cell types [25].

## 3. Results

### 3.1. MR Analysis Results of pQTL on PF

The MR analysis results, shown in the Figure 2, reveal that 64 genetic loci are significantly associated with PF. For instance, the CRP locus (*p* = 0.005, OR = 1.132, CI [1.084, 1.589]) and the ARL1 locus (*p* = 0.008, OR = 1.683, CI [1.147, 2.468]) demonstrate a positive correlation with PF. Conversely, the ICOSLG locus (*p* = 0.006, OR = 0.440, CI [0.245, 0.789]) and the AGER locus (*p* = 0.007, OR = 0.847, CI [0.751, 0.957]) show a negative correlation with PF, implying that these proteins may play a protective role in inhibiting or reversing PF. These findings suggest that alterations in the expression levels of specific proteins could serve as potential biomarkers influencing variations in PF.

The reverse Mendelian analysis reveals a positive causal relationship between PF and *NPTX1* (*p* = 0.038, OR = 1.020, CI [1.001, 1.040]), *IL31* (*p* = 0.040, OR = 1.019, CI [1.000, 1.038]), and *CTSE* (*p* = 0.030, OR = 1.022, CI [1.002, 1.042]). This suggests that PF may induce an increase in the expression of *NPTX1*, *IL31*, and *CTSE*, potentially enhancing local immune responses and accelerating the progression of the disease.

### 3.2. Enrichment Analysis

The GO enrichment analysis includes three categories: BP, CC, and MF, with the goal of identifying the functional characteristics of genes and the biological processes in which they are involved. The KEGG enrichment analysis aids in understanding the role of genes within specific biological pathways [26]. As shown in the Figure 3, the BP category includes significantly enriched terms related to bone growth, endochondral bone growth, bone development, and bone mineralization. In the CC category, the enrichment of extracellular matrix components, membrane microdomains, and synaptic vesicles highlights active cellular interactions, cell membrane structure, and neurobiological functions. Additionally, in the MF category, enriched terms such as S100 protein binding, cell adhesion mediator activity, and SNARE binding underscore the importance of gene products in biochemical functions like cell signaling, membrane fusion, and protein binding. The top five pathways based on KEGG enrichment analysis, as shown in the Figure 4, are the PI3K/Akt signaling pathway, focal adhesion, human papillomavirus infection, ECM–receptor interaction, and cytokine–cytokine receptor interaction.

### 3.3. PPI Network and Core Gene Screening

The 64 gene proteins were submitted to the STRING database for network construction. The resulting files were imported into Cytoscape for visualization, and core genes were identified using the cytoHubba plugin. As shown in Figure 5, the PPI network includes 59 nodes and 232 edges. The core genes identified include *CDH1*, *CRP*, *VTN*, *COL1A1*, *MAPK8*, *TGFB3*, *TIMP3*, *COMP*, *LYN*, and *EZR*. Network analysis revealed that these genes are involved in biological processes such as cell signaling, cell adhesion, ECM remodeling, and immune response. These processes are associated with fibroblast activation, abnormal ECM accumulation, and chronic inflammation, aligning with the fundamental pathogenesis of PF [27].

### 3.4. Prediction of Candidate Drugs

In this study, we utilized DSigDB to predict potential intervention drugs and conducted drug enrichment analysis using the R package clusterProfiler. The results revealed that sorafenib, ascorbic acid (Vitamin C), (E)-4-hydroxy-2-nonenal (2-nonenal, 4-hydroxy-(2E,4R)-), alpha-tocopherol (Vitamin E), and the mitogen-activated protein kinase kinase inhibitor (PD 98059) are key drugs associated with core genes (Figure 6 and Figure 7).

### 3.5. Molecular Docking

To evaluate the binding affinity of the candidate drugs with their targets and assess the druggability of these targets, molecular docking was performed in this study. We used the CB-Dock2 analysis platform to investigate the binding between the five candidate drugs and their corresponding target proteins (Figure 8). In this analysis, we observed that the binding energy between Sorafenib and *MAPK8* was the lowest (−10.6 kcal/mol), indicating a highly stable binding interaction. Overall, the binding energies ranged from −4.4 kcal/mol to −10.6 kcal/mol (Table 1), with each drug showing good binding affinity to its corresponding target protein.

### 3.6. Single-Cell Sequencing

In this study, we analyzed the expression of core genes in lung tissue cells using the PanglaoDB platform, selecting datasets related to human lung proximal airway mesenchymal cells for single-cell RNA sequencing analysis. The results showed that the *CDH1* gene was expressed in basal cells, luminal epithelial cells, and type II alveolar cells, with higher expression levels observed in type II alveolar cells. The *VTN* and *COL1A1* genes were highly expressed in fibroblasts, suggesting their key roles in fibroblast function. The *MAPK8* gene was expressed in fibroblasts, type II alveolar cells, and basal cells, with the highest expression observed in fibroblasts. The *TGFB3* gene was significantly overexpressed in fibroblasts. The *TIMP3* gene was expressed in various cell types, with the highest expression observed in fibroblasts. The *COMP* gene also showed high expression levels in fibroblasts. The *LYN* gene was expressed in all cell types, with the most prominent expression observed in type II alveolar cells. The *EZR* gene was significantly expressed in various cell types, including type II alveolar cells, ependymal cells, and smooth muscle cells (Figure 9 and Figure 10).

These results further elucidate the roles of these genes in the pathogenesis of PF and provide valuable insights and theoretical support. First, most potential targets (such as *VTN*, *COL1A1*, *MAPK8*, *TGFB3*, *TIMP3*, and *COMP*) were highly expressed in fibroblasts, suggesting their key roles in fibroblast activation, proliferation, and extracellular matrix (ECM) remodeling, which contribute to fibrotic tissue accumulation and disease progression. Second, certain genes (such as *CDH1*, *LYN*, and *EZR*) were significantly expressed in type II alveolar cells and other epithelial cells, indicating their possible involvement in epithelial–mesenchymal transition (EMT), epithelial barrier maintenance, and injury repair. This provides new research directions for understanding the relationship between lung epithelial damage, repair mechanisms, and fibrosis. Moreover, the high expression of *EZR* in multiple cell types suggests its potential multifunctionality in cell–cell interactions, mechanosignaling, and the regulation of various signaling pathways.

## 4. Discussion

Under specific conditions, the plasma proteome exhibits notable stability, significant variability, and predictability, making it a critical resource for PF research [28,29]. Despite individual differences, statistical methods in large-scale studies minimize variability, facilitating the identification of key PF-associated proteins. These findings are crucial for improving our understanding and diagnosis of PF. This study applied the MR analysis method and identified 64 potential therapeutic targets in the plasma proteome linked to PF. To explore the biological relevance of these targets, we conducted enrichment analysis and PPI network analysis to identify core genes, as well as drug prediction and molecular docking. Single-cell sequencing further validated the therapeutic potential of these genes. The results indicate that most predicted drugs play roles in anti-inflammatory, antioxidant, and hormone regulation processes, mechanisms essential for PF treatment. These findings expand the clinical applications of existing drugs and provide a theoretical basis for their future use in PF treatment.

Regarding the identified core genes, *CDH1* produces E-cadherin through transcription and translation, maintaining cell–cell adhesion and epithelial phenotype stability in epithelial cells. The downregulation of E-cadherin expression is a hallmark of epithelial-to-mesenchymal transition (EMT), a key molecular mechanism in the pathogenesis of PF [30]. EMT promotes the transformation of epithelial cells into mesenchymal cells, enhancing their migratory ability and secretion of fibrotic factors, which leads to the excessive accumulation of fibrous tissue [31]. *CRP*, an acute-phase reactant protein, plays an important role in the pathogenesis and disease progression monitoring of PF. Elevated *CRP* levels are closely associated with the onset, progression, and prognosis of PF [32]. It directly participates in the inflammatory response and fibrosis process in lung tissue by activating the complement system, modulating immune cell functions, promoting macrophage and fibroblast activation, enhancing cytokine secretion, and accelerating the development of fibrosis [33]. *VTN* is a multifunctional glycoprotein that regulates fibroblast migration and activation through interactions with integrins and other cell surface receptors. It also promotes the activation of the TGF-β signaling pathway, driving EMT and collagen deposition, which accelerates lung tissue fibrosis [33]. The *COL1A1* gene encodes the α1 chain of type I collagen, a major structural component of the extracellular matrix (ECM). The continuous activation of pro-fibrotic signaling pathways induces the overexpression of *COL1A1*, accelerating fibrosis progression [34]. The upregulation of *COL1A1* is negatively correlated with the severity, prognosis, and survival rates of PF patients. Inhibiting its expression or blocking related pathways may help to slow the excessive deposition of type I collagen, offering potential intervention strategies for PF treatment [35]. *MAPK8*, a key member of the MAP kinase signaling pathway, participates in cell stress responses, proliferation, differentiation, and apoptosis. In response to inflammatory stimuli, it promotes the release of inflammatory mediators, exacerbating lung tissue inflammation [36]. *MAPK8* also regulates the TGF-β signaling pathway, promoting EMT, increasing fibrotic factor deposition, accelerating fibrous tissue formation, and driving the progression of fibrosis through its impact on fibroblast proliferation and activation [37,38]. *TGFB3*, a member of the TGF-β family, plays a role in cell proliferation, differentiation, migration, and ECM remodeling [39]. It activates both Smad-dependent and independent signaling pathways to promote fibroblast proliferation and excessive ECM protein deposition, leading to lung tissue fibrosis [40]. *TGFB3* also induces EMT, transforming epithelial cells into mesenchymal cells, further exacerbating fibrosis progression. *TIMP3*, a matrix metalloproteinase (MMP) inhibitor, regulates ECM homeostasis and tissue remodeling. The abnormal expression (either upregulation or downregulation) of *TIMP3* is commonly observed in PF patients [41]. Overexpression leads to ECM accumulation, promoting the formation of fibrous tissue. *COMP* (cartilage oligomeric matrix protein) interacts with collagen and other matrix proteins to promote ECM accumulation and stabilization, thereby exacerbating lung tissue fibrosis [42]. It affects fibroblast proliferation and migration, regulates the activity of pro-fibrotic factors, drives EMT, activates fibroblasts, and enhances fibrous tissue formation [43]. *LYN*, a Src family tyrosine kinase, activates the TGF-β/Smad signaling pathway, promoting fibroblast proliferation and collagen synthesis, which accelerates ECM accumulation and tissue fibrosis [44]. *LYN* modulates immune cell function, affecting the activation and differentiation of macrophages and T cells, thus fostering a chronic inflammatory environment [45]. It further promotes the expression of fibrotic genes and the formation of fibrous tissue through pathways such as PI3K/Akt and NF-κB [46]. The *EZR* gene encodes the ezrin protein, which links the cell membrane to the cytoskeleton and regulates cell morphology, migration, and signal transduction [47]. *EZR* promotes epithelial cell migration and EMT, activates fibroblast proliferation, and accelerates excessive ECM deposition, leading to the destruction of lung tissue structure and dysfunction [48].

The PI3K/Akt signaling pathway directly accelerates fibrosis by promoting fibroblast proliferation, anti-apoptosis, and differentiation into myofibroblasts [49]. Focal adhesion and ECM–receptor interactions also play a crucial role in cell–matrix interactions, promoting cell migration and ECM accumulation, thereby driving the progression of PF [50]. Immune factors in cytokine–cytokine receptor interactions, such as TGF-β, promote the sustained progression of PF by activating fibroblasts and collagen synthesis. Although human papillomavirus infection is primarily associated with cancer, its induced chronic immune response may indirectly contribute to PF by disrupting immune responses and increasing inflammation, thereby accelerating fibrosis. Therefore, these pathways collectively contribute to PF by regulating immune responses, cell migration and proliferation, and ECM remodeling, offering potential therapeutic targets for treatment [51].

Sorafenib, vitamin C, and vitamin E have attracted significant attention as potential therapeutic agents for PF. Sorafenib, a multi-target kinase inhibitor, inhibits fibroblast activation and collagen deposition by targeting the MAPK/ERK signaling pathway [52]. Preclinical studies show that sorafenib effectively reverses the expression of fibrosis-related genes and modulates key biological pathways in the TGF-β1-induced pulmonary fibrosis rat model, particularly those involved in inflammation, immune response, and tissue remodeling, underscoring its potential as an effective therapeutic intervention for pulmonary fibrosis [53]. The anti-fibrotic effects of vitamin C and vitamin E, as classical antioxidants, have been confirmed in multiple studies. Vitamin C mitigates alveolar epithelial cell damage by inhibiting oxidative stress pathways and suppresses fibroblast transformation into myofibroblasts [54,55]. Vitamin E improves lung tissue structure and function by reducing TGF-β levels and inhibiting the expression of fibrosis-related markers, such as α-SMA and type I collagen [56]. Further studies indicate that vitamin E significantly slows the progression of bleomycin-induced pulmonary fibrosis by improving iron metabolism and mitochondrial function [57]. Clinical studies have also investigated the combined supplementation of vitamins C, E, and D in patients with IPF. The results showed that supplementation with these vitamins significantly improved lung function in patients, particularly in respiratory parameters such as FEV1, IRV, and RV, while also significantly reducing TGF-β and inflammatory markers like hs-CRP [58]. Molecular docking results indicate that sorafenib has the lowest binding energy with MAPK8 (−10.6 kcal/mol), consistent with the previously identified mechanism of sorafenib in regulating the MAPK pathway. Furthermore, vitamins C and E demonstrate stable binding with targets, with binding energies ranging from −4.4 kcal/mol to −6.8 kcal/mol, further supporting their potential intervention in PF through antioxidant and anti-inflammatory pathways.

In conclusion, the drug targets identified in this study are closely associated with PF and demonstrate substantial therapeutic potential. These findings support the design of PF treatments that target specific molecules, providing a critical theoretical and practical foundation for the development of future therapies.

Despite the significant progress achieved in this study, several limitations persist. First, the study population mainly comprises individuals of European descent, which may limit the generalizability of the findings. Further research involving diverse ethnic groups is required to confirm the broader applicability of these findings. This study did not include a subtype stratification analysis of PF, which may have introduced a certain degree of confounding bias [59]. However, the elevated expression of core genes, such as COL1A1 and MAPK8, in fibroblasts and their strong association with the TGF-β signaling pathway [60] are closely linked to the pathological features of IPF. These findings suggest that these genes play a pivotal role in the pathological mechanisms underlying IPF. Future studies should validate the specificity of these targets in well-defined patient subgroups and further investigate their regulatory differences across various PF subtypes. In addition, while this study did not incorporate subtype stratification analysis, single-cell RNA sequencing revealed that the elevated expression of core genes, including CDH1, VTN, and COL1A1, in fibroblasts and type II alveolar epithelial cells is strongly associated with the molecular characteristics of the UIP pattern [61,62]. These findings imply that these genes may have a shared pathogenic role across diverse subtypes of PF [63]. Future research should integrate subtype classification data to further elucidate the specific functions and underlying mechanisms of these genes in different pathological contexts. Second, despite efforts to minimize bias, MR analysis may still be affected by unmeasured confounders or pleiotropy. In this study, we assessed the robustness of our findings through various sensitivity analyses, including MR-Egger regression, weighted median and mode methods, Cochran’s Q test, and the MR-PRESSO pleiotropy test. However, we acknowledge the limitation of not validating these findings against independent datasets, such as the UK Biobank. Replication analyses using independent pQTL datasets would further strengthen the reliability of our findings. Future research should emphasize integrating resource-rich, high-quality datasets to thoroughly validate our preliminary findings. Furthermore, the accuracy of molecular docking is dependent on the quality of both the protein structure and the ligand. Moreover, limitations in single-cell sequencing, such as dataset selection and analytical platforms, may result in an incomplete representation of the phenotypic and functional characteristics of specific cell types or subpopulations. Although potential drug targets were identified, their clinical efficacy remains uncertain and needs validation through experimental studies and clinical trials. Finally, chronic kidney disease (CKD) may substantially alter the composition of the plasma proteome [64,65]. Since this study did not account for CKD in the data screening process, the results could be affected. Addressing these limitations will enhance future research and improve our understanding of PF and its therapeutic strategies.

Based on the research findings and their limitations, we propose several recommendations for expanding future research directions. First, to enhance the generalizability of the results, study populations from diverse ethnicities and regions should be included to verify the applicability of the findings to a broader patient cohort. Additionally, future research could integrate multi-omics data and environmental factors to explore the molecular mechanisms of PF from a multidimensional perspective. Moreover, higher-resolution structural biology techniques and molecular dynamics simulations could improve the accuracy of molecular docking analyses. Additionally, incorporating spatial transcriptomics and multi-omics single-cell technologies would enhance the comprehensiveness and precision of data interpretation, better revealing the functional characteristics of various cell subpopulations in PF. Future studies should also validate the identified targets in laboratory and animal models, as well as in early clinical trials, to assess their clinical applicability and safety. Furthermore, kidney-function-related parameters should be incorporated into the study design, and CKD patients should be excluded to minimize potential confounding effects on the results. In summary, by implementing the above strategies, future research is expected to deepen our understanding of the pathogenesis of PF, expand potential therapeutic interventions, and ultimately improve patient prognosis.

## 5. Conclusions

This study systematically evaluated the causal relationship between plasma proteins and PF by integrating MR analysis, identifying 64 potential therapeutic targets related to PF and screening core genes. Additionally, reverse Mendelian randomization analysis identified three key genes associated with PF, offering a novel perspective on the bidirectional causal relationship between plasma proteins and PF. The results suggest that these key genes play vital roles in ECM remodeling, inflammatory response, and EMT, with significant expression primarily in fibroblasts and type II alveolar epithelial cells. Through drug enrichment and molecular docking analysis, the study identified candidate drugs, including sorafenib, vitamin C, and vitamin E, which exhibited significant therapeutic potential in anti-inflammatory, antioxidant, and regulatory signaling pathways. The findings of this study offer valuable insights for PF treatment, with the potential to reduce drug development costs and support the formulation of personalized medical plans. These results are crucial for advancing the understanding of the molecular mechanisms of PF and developing precise therapeutic strategies targeting core genes. However, further experimental studies and clinical trials are required to validate the therapeutic potential of drugs targeting these proteins and confirm their safety and efficacy in clinical settings. In summary, this study offers new theoretical insights and practical evidence for PF treatment, providing a solid foundation for future research and drug development.

## Figures and Tables

**Figure 1 biology-14-00200-f001:**
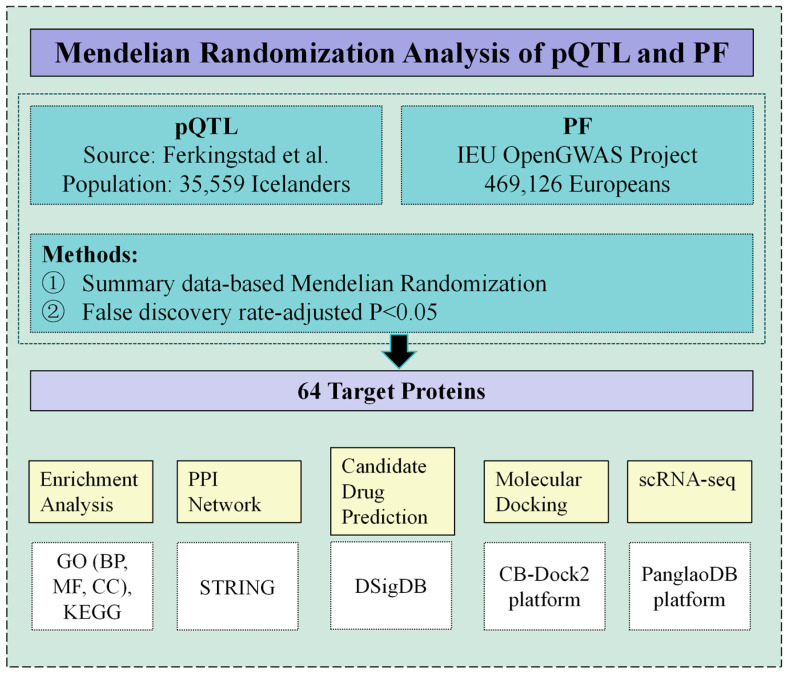
Overview of the study design [13].

**Figure 2 biology-14-00200-f002:**
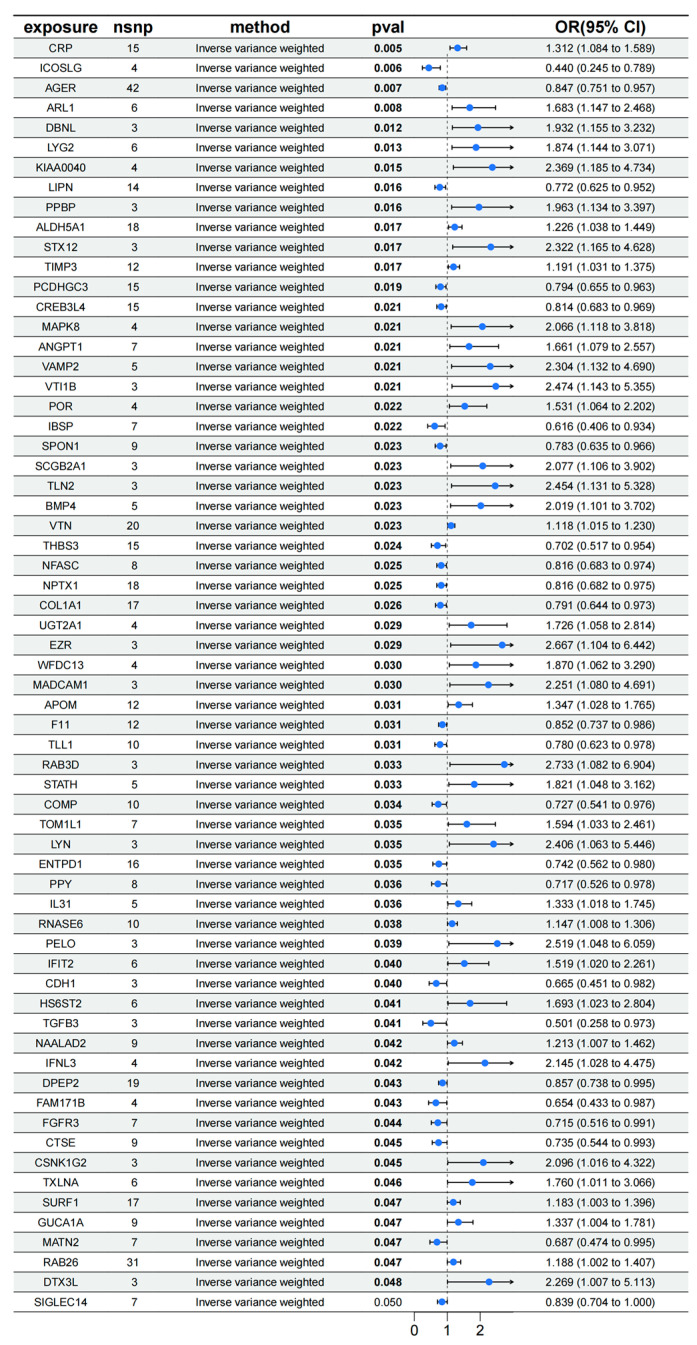
MR analysis results of pQTL on PF.

**Figure 3 biology-14-00200-f003:**
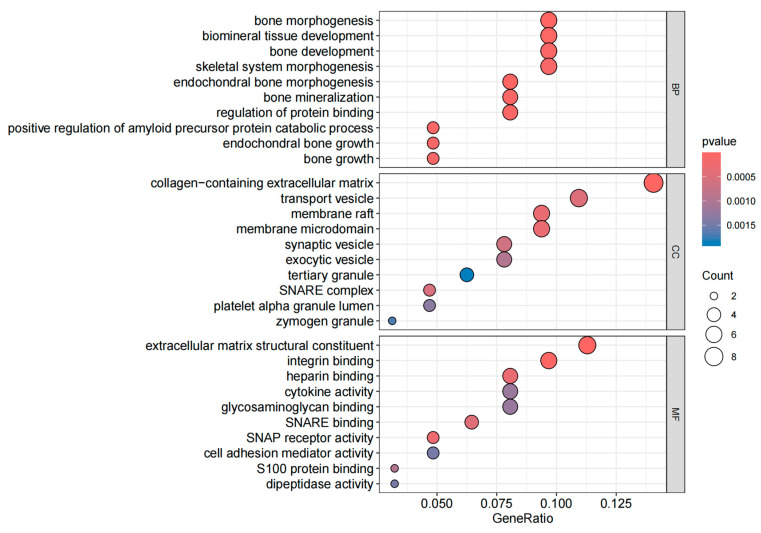
GO enrichment results for three terms.

**Figure 4 biology-14-00200-f004:**
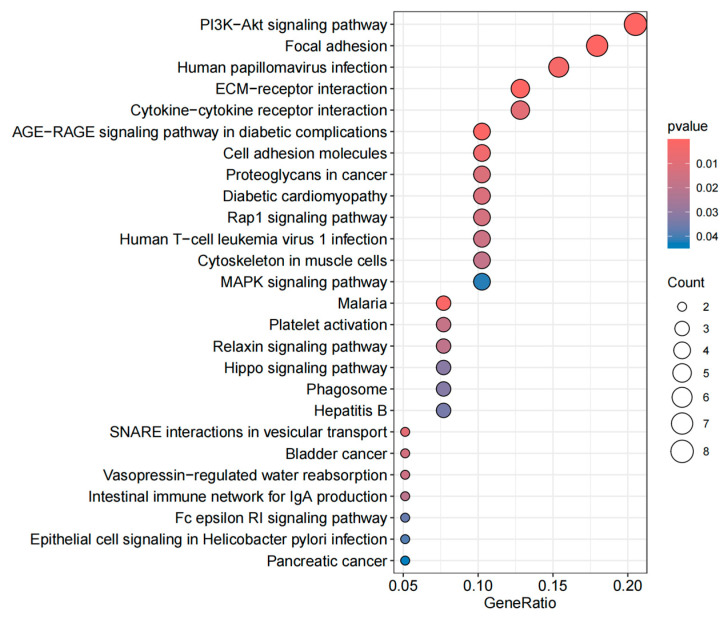
KEGG enrichment results.

**Figure 5 biology-14-00200-f005:**
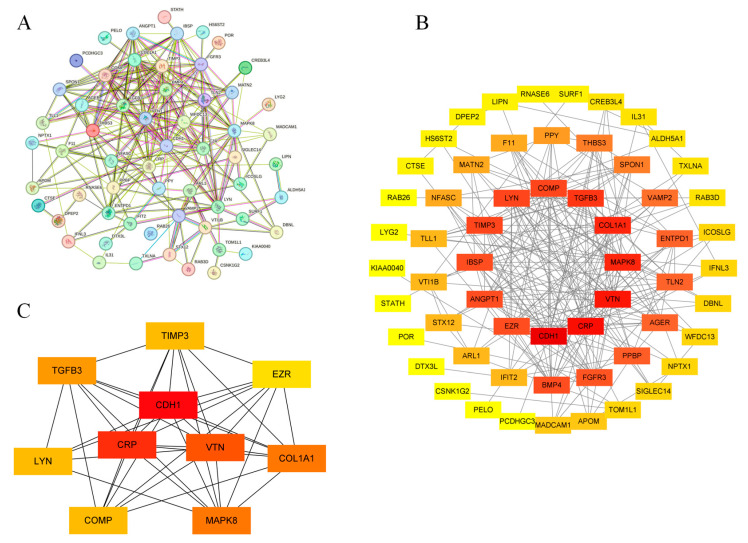
PPI network construction diagram. (**A**): PPI network built with STRING. (**B**): Full PPI network of selected genes. Key clusters with hub genes highlighted in red. (**C**): Core sub-network showing interactions among top hub genes.

**Figure 6 biology-14-00200-f006:**
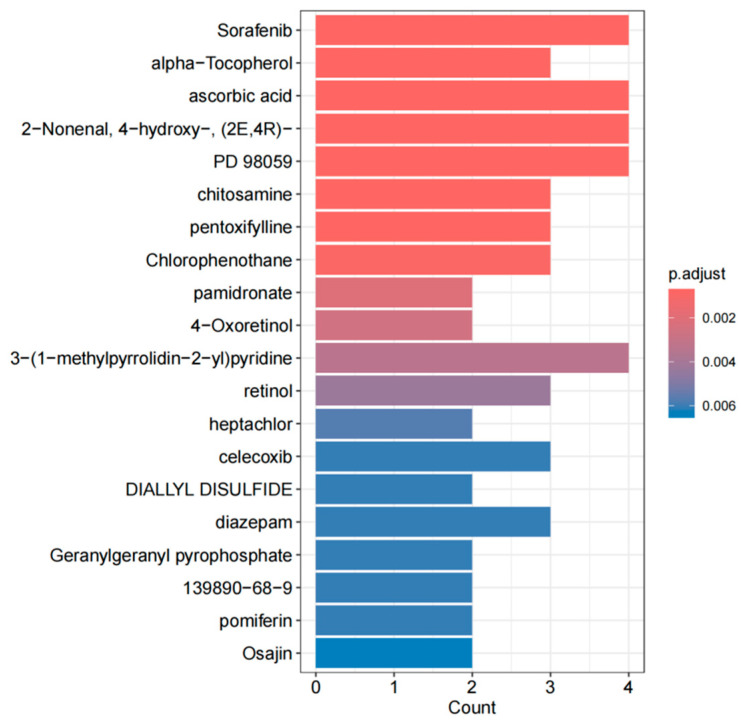
Bar chart of drug prediction results.

**Figure 7 biology-14-00200-f007:**
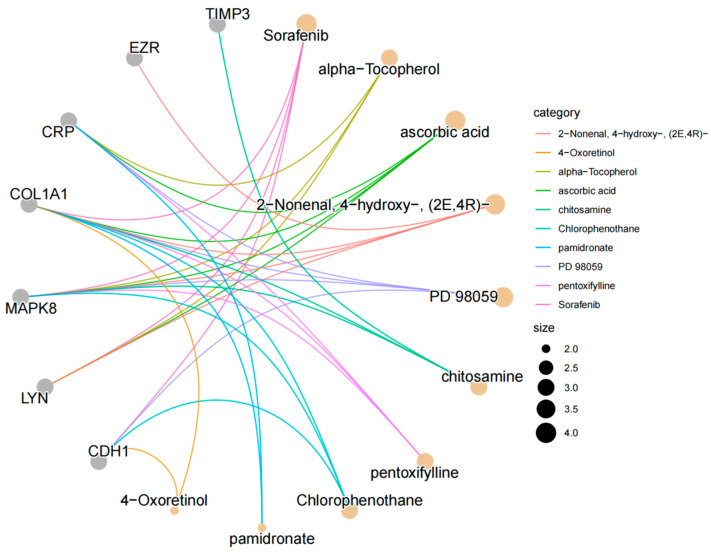
Gene–drug interaction network diagram.

**Figure 8 biology-14-00200-f008:**
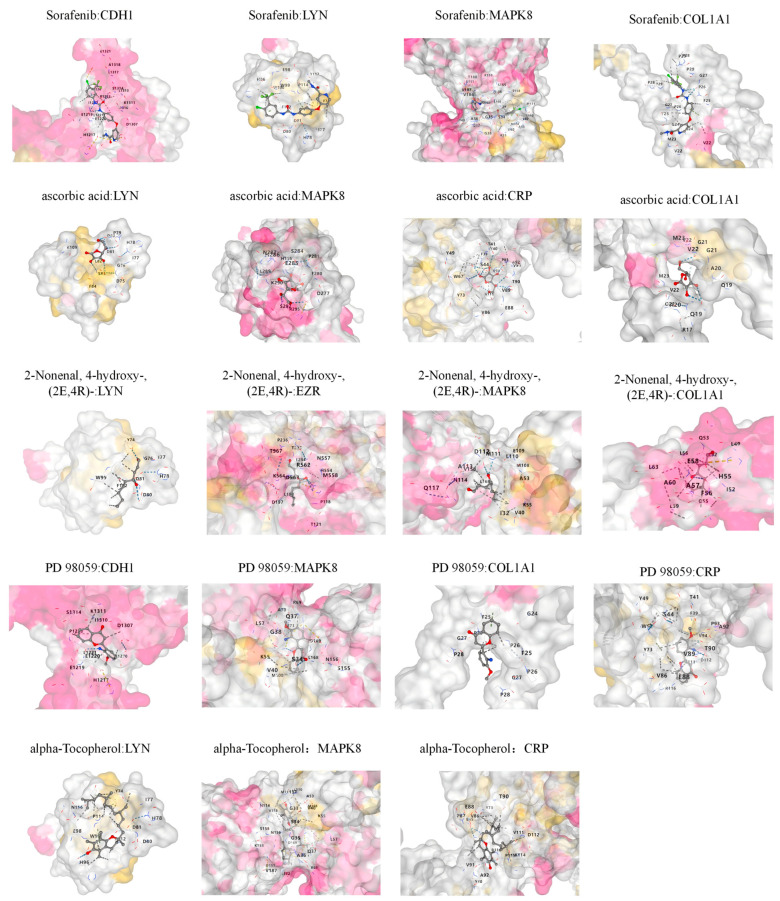
Docking results of available proteins small molecules.

**Figure 9 biology-14-00200-f009:**
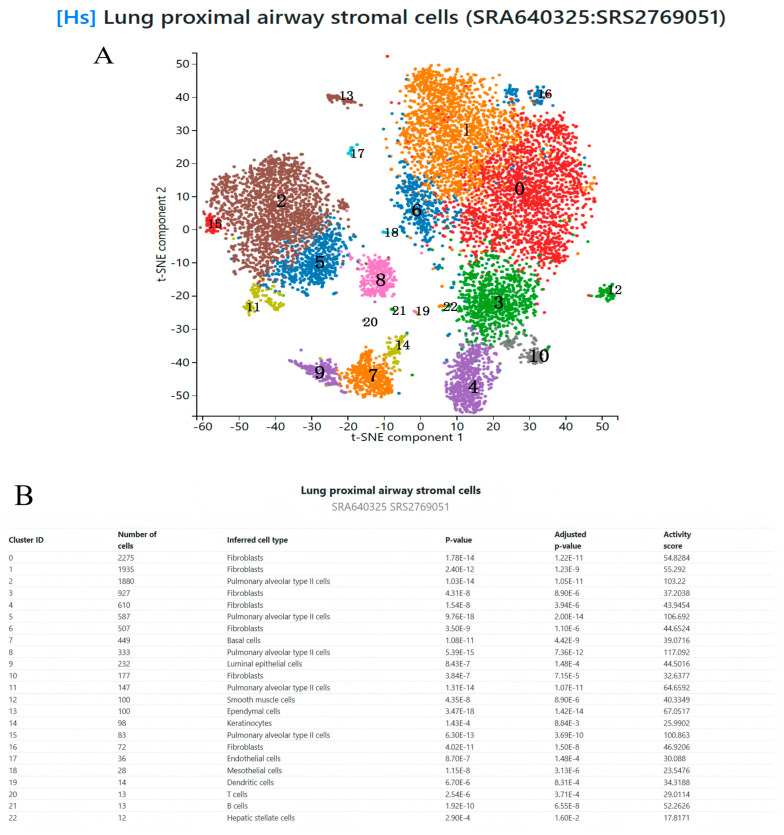
(**A**): t-SNE cell distribution plot. Based on single-cell RNA sequencing data, the t-SNE dimensionality reduction plot illustrates the heterogeneity of proximal airway stromal cells in the lung. Different colors represent cell subpopulations. (**B**): Key information of 22 cell clusters.

**Figure 10 biology-14-00200-f010:**
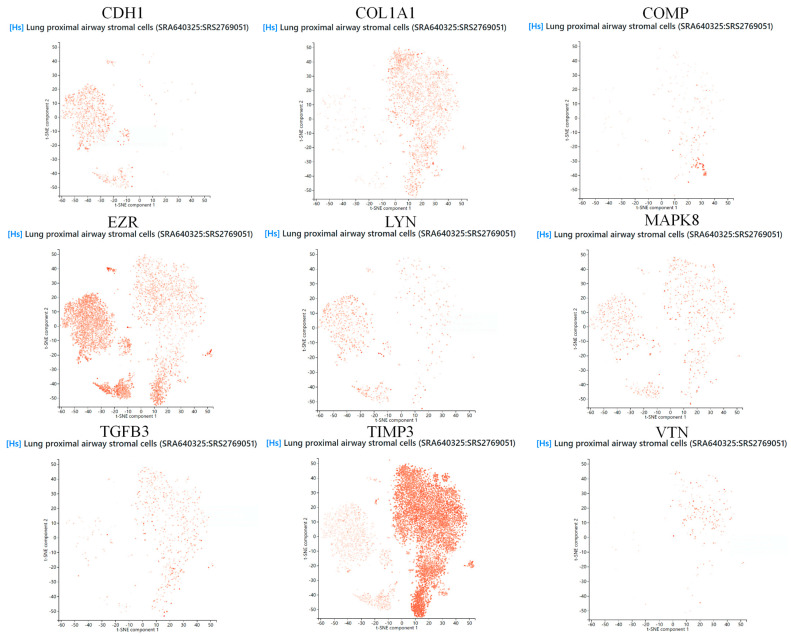
Scatterplots showing the expression distribution of core genes across clusters in lung proximal airway stromal cells.

**Table 1 biology-14-00200-t001:** Docking results of available proteins with small molecules.

Drug	PubChem ID	Target	UniProt ID	Binding Energy (kcal/mol)
Sorafenib	216239	*CDH1*	P12830	−8.5
Sorafenib	216239	*LYN*	P07948	−7.1
Sorafenib	216239	*MAPK8*	P45983	−10.6
Sorafenib	216239	*COL1A1*	P02452	−8.8
Alpha-tocopherol	14985	*LYN*	P07948	−5.7
Alpha-tocopherol	14985	*MAPK8*	P45983	−6.8
Alpha-tocopherol	14985	*CRP*	P02741	−6.2
Ascorbic acid	54670067	*LYN*	P07948	−4.4
Ascorbic acid	54670067	*MAPK8*	P45983	−5.9
Ascorbic acid	54670067	*COL1A1*	P02452	−5.0
Ascorbic acid	54670067	*CRP*	P02741	−5.7
2-nonenal, 4-hydroxy-, (2E,4R)-	11957428	*LYN*	P07948	−4.6
2-nonenal, 4-hydroxy-, (2E,4R)-	11957428	*EZR*	P15311	−5.0
2-nonenal, 4-hydroxy-, (2E,4R)-	11957428	*MAPK8*	P45983	−5.0
2-nonenal, 4-hydroxy-, (2E,4R)-	11957428	*COL1A1*	P02452	−5.4
PD 98059	4713	*CDH1*	P12830	−6.6
PD 98059	4713	*MAPK8*	P45983	−8.2
PD 98059	4713	*COL1A1*	P02452	−6.9
PD 98059	4713	*CRP*	P02741	−7.4

## Data Availability

All the data used in this study had been publicly available.

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
