# Peer review of "Assessing the Causal Relationship Between Plasma Proteins and Pulmonary Fibrosis: A Systematic Analysis Based on Mendelian Randomization"

_biology, 2025, doi:10.3390/biology14020200_

Round 1

Reviewer 1 Report

Comments and Suggestions for Authors

The authors of this study performed an MR analysis integrating Pulmonary Fibrosis GWAS and pQTL data. Their analysis identified 64 significant causal associations, however it is not clear if this number is after multiple test corrections. And then they performed gene set enrichment, drug prediction and single-cell analyses. I found the study interesting, however there are some points that need further explanation:

Main comments:

The sample size described in ref 13 is much smaller than the sample size reported by the authors. Could the authors explain this discrepancy?

Can the authors explain why they used SMR in the initial analysis and TwoSampleMR in the reverse? If they used SMR for the initial analysis, did they perform the HEIDI test?

From the results it seems that they did not use SMR but TwoSampleMR for all the analyses. Given that they described that they performed: weighted median method, simple mode method, weighted mode method, and MR Egger, Cochrane’s Q test and random effects IVW, and pleiotropy test through MR-PRESSO, but these results are not available to the reader.

I believe a replication analysis using an independent pQTL dataset, such as UKB could help to make the initial findings more reliable. Especially given that the authors did not report any of the other tests.

In section 3.1 it is not clear if the authors perform multiple test corrections, similarly to figures 3 and 4. If the p-values are adjusted it should be clear on the figures and the results presentation.

Minor comments:

Lines 240-251, 275-282,  are mostly a discussion of the results rather than the actual presentation of them. 

The graphics in figure 9 are two small to read.

There is a typo in line 255.

Author Response

Dear reviewer:

I sincerely thank you for your careful review of this paper and your valuable comments! Your questions have greatly enhanced the rigour and readability of the study. The following is a list of specific revisions in response to each of your questions:

Main comments:

1.Q: The sample size described in ref 13 is much smaller than the sample size reported by the authors. Could the authors explain this discrepancy?

A: We sincerely thank the reviewer for pointing out this discrepancy. Upon careful review, we identified an error in the original citation of reference 13. We have now corrected the reference and explicitly clarified the data source (Page 3, Section 2.1.2, Lines 128-131). Additionally, we have provided a direct link to the original dataset in the deCODE database to ensure transparency and accuracy. These revisions enhance the clarity and reliability of our data presentation.

2.Q: Can the authors explain why they used SMR in the initial analysis and TwoSampleMR in the reverse? If they used SMR for the initial analysis, did they perform the HEIDI test?

A: We greatly appreciate the reviewer’s keen observation regarding the methodological description. The mention of "SMR method" in the original manuscript was an error. In fact, all analyses were conducted using the TwoSampleMR package for bidirectional Mendelian randomization (MR) analysis. We have thoroughly revised the "Mendelian Randomization Analysis" section (Page 3, Section 2.1.3, Lines 142-168) to provide a detailed description of the bidirectional MR analysis steps using TwoSampleMR. This includes comprehensive explanations of the Inverse-Variance Weighted (IVW) method, MR-Egger regression, weighted median method, and other relevant analyses. These revisions ensure the methodological accuracy and clarity of our work. We sincerely thank the reviewer for their meticulous review and valuable feedback.

3.Q: From the results it seems that they did not use SMR but TwoSampleMR for all the analyses. Given that they described that they performed:  weighted median method, simple mode method, weighted mode method, and MR Egger, Cochrane’s Q test and random effects IVW, and pleiotropy test through MR-PRESSO, but these results are not available to the reader.

A: We sincerely thank the reviewer for this valuable suggestion. In response, we have now included the results of multiple testing corrections in Supplementary File 1. Additionally, we have added visualizations of these results to the supplementary materials (Page 3, Section 2.1.3, Lines 157-168). These updates ensure that all analytical outcomes are transparent and accessible to readers, enhancing the reproducibility and clarity of our findings.

4.Q;I believe a replication analysis using an independent pQTL dataset, such as UKB could help to make the initial findings more reliable. Especially given that the authors did not report any of the other tests.

A: We sincerely appreciate the reviewer’s insightful suggestion. While we acknowledge the importance of validating our findings using an independent dataset such as the UK Biobank, current data availability limitations prevent us from conducting such analyses at this stage. However, we have addressed this limitation in the revised manuscript (Discussion, Page 18, Paragraph 4, Lines 478-486), explicitly stating our intention to incorporate independent cohorts like the UK Biobank in future studies to further validate our results. This addition strengthens the transparency and scientific rigor of our work.

5.Q: In section 3.1 it is not clear if the authors perform multiple test corrections, similarly to figures 3 and 4. If the p-values are adjusted it should be clear on the figures and the results presentation.

A: We sincerely thank the reviewer for this important reminder. For Figures 3 and 4, the enrichment analysis was designed to illustrate potential biological pathways rather than to serve as strict validation. Therefore, the figures display uncorrected p-values. However, we have explicitly described the correction methods in the main text (Page 5, Section 2.2, Lines 198-205) and provided detailed results in Supplementary File 2. This ensures the transparency and completeness of our data.

Minor comments:

1.Q: Lines 240-251, 275-282,  are mostly a discussion of the results rather than the actual presentation of them. 

A: We sincerely thank the reviewer for this valuable suggestion. In response, we have restructured the relevant paragraphs to ensure that the Results section focuses solely on the objective presentation of data. All discussion-oriented content has been moved to the Discussion section (Page 18, Paragraphs 1 and 2, Lines 439-456 and 464-476). This revision improves the clarity and organization of the manuscript, ensuring that each section fulfills its intended purpose.

2.Q: The graphics in figure 9 are two small to read.

A: We sincerely thank the reviewer for this constructive suggestion. In response, we have redesigned Figure 9 and added a new Figure 10 (Page 7). The revised Figure 9 (A) now clearly displays the t-SNE dimensionality reduction results, while (B) highlights the expression patterns of core genes. To improve readability, we have removed redundant cluster labels and increased the size of the scatter plots. High-resolution versions of these figures are available in the supplementary materials (Pages 14-16). These modifications enhance the clarity and interpretability of the visual data.

3.Q: There is a typo in line 255.

A: We sincerely thank the reviewer for identifying this typographical error. We have carefully reviewed the manuscript and corrected all spelling and grammatical errors to ensure the language is clear, accurate, and polished. These revisions have significantly improved the overall quality of the manuscript, and we are grateful for the reviewer’s meticulous attention to detail.

Reviewer 2 Report

Comments and Suggestions for Authors

Thanks for the opportunity to review the manuscript by Moxuan Han and colleagues. The authors evaluated the causal relationship between plasma proteins and PF by integrating SMR analysis. They identified 64 potential therapeutic targets related to PF and screening core genes. Also, reverse MR analysis identified three key genes associated with PF and proposed them as a novel perspective on the bidirectional causal relationship between plasma proteins and PF.  According to the results, they suggested that these key genes play vital roles in ECM remodeling, inflammatory response, and EMT, with significant expression primarily in fibroblasts and type II alveolar epithelial cells. Finally, the drug enrichment and molecular docking analysis identified candidate drugs that showed therapeutic potential in anti-inflammatory, antioxidant, and regulatory signaling pathways. 

The manuscript is really interesting and well-written in general. I have some minor comments for the authors to be considered.

From the introduction section, the authors should consider that "pulmonary fibrosis" is a basket term that groups several entities. Then, some worse outcomes have been identified in idiopathic pulmonary fibrosis than in non-idiopathic phenotypes. Please include a brief paragraph about this point. Also, the authors should consider the histopathologic and CT scan (tomographic) pattern. There are clear differences across the UIP or NSIP patterns and molecular conditions.

Both issues should be considered in the general description of the clinical and demographic characteristics of included subjects, which is absent in the present version. They should also be addressed in the methods and results sections. 

Minor comments:

In the 2.1.2. Outcome Data section, please expand the information about PF-related genetic data. The current status is vague and imprecise. Describe the variables considered as related to PF.

Please use the gene symbols, including italics, to differentiate genes from proteins properly throughout the manuscript.

Please review the redation throughout the manuscript; there are some minor typos—for example, lines 255, 265, etc.

Author Response

We thank the reviewer for the thorough evaluation of our study and the constructive feedback. We have carefully read each comment and made revisions based on your suggestions. Below are the detailed explanations of the revisions:

1.Q:  From the introduction section, the authors should consider that "pulmonary fibrosis" is a basket term that groups several entities. Then, some worse outcomes have been identified in idiopathic pulmonary fibrosis than in non-idiopathic phenotypes. Please include a brief paragraph about this point. Also, the authors should consider the histopathologic and CT scan (tomographic) pattern. There are clear differences across the UIP or NSIP patterns and molecular conditions.

Both issues should be considered in the general description of the clinical and demographic characteristics of included subjects, which is absent in the present version. They should also be addressed in the methods and results sections. 

A: We appreciate the reviewers' constructive suggestions regarding the Introduction section. As recommended, we have incorporated an expanded discussion on pulmonary fibrosis heterogeneity in the Introduction (Page 2, Paragraph 1, Lines 61-69). This addition specifically addresses the distinct clinical outcomes between idiopathic pulmonary fibrosis (IPF) and non-IPF subtypes, along with differential histopathological and CT imaging patterns of UIP and NSIP. These enhancements better contextualize the study's rationale and clinical relevance.

The methodological rationale for not subtyping pulmonary fibrosis cases is explicitly detailed in Section 2.1.2 (Page 4, Lines 131-136). Subsequent discussion of this methodological consideration appears in Section 4 (Page 18, Paragraph 4, Lines 471-475), where we acknowledge current limitations and propose future research directions to address this aspect through more comprehensive subtype analyses.

2.Q: In the 2.1.2. Outcome Data section, please expand the information about PF-related genetic data. The current status is vague and imprecise. Describe the variables considered as related to PF.

A: We greatly appreciate your valuable comments concerning the clarification of data sources. In response, we have expanded the Methods section (Page 4, Section 2.1.2, Lines 128-131) to include detailed information about the PF-related genetic data. This addition specifies the data sources, database accession numbers, and original literature references, thereby ensuring both the accuracy and transparency of our data provenance.

3.Q: Please use the gene symbols, including italics, to differentiate genes from proteins properly throughout the manuscript.

A: Following your recommendation, we have meticulously reviewed the entire manuscript to ensure proper formatting of gene symbols in italics and protein names in standard font. These formatting adjustments have been consistently applied throughout the manuscript to comply with established academic conventions.

4.Q: Please review the redation throughout the manuscript; there are some minor typos—for example, lines 255, 265, etc.

A: We have carefully corrected all spelling errors identified in the manuscript. We sincerely appreciate your valuable feedback, which has significantly contributed to enhancing the overall quality of our work.

Round 2

Reviewer 2 Report

Comments and Suggestions for Authors

Thanks for attending to my previous concerns.

Author Response

Thank you once again for your thoughtful consideration of the points I raised earlier. I truly appreciate your time and effort in addressing these issues, and I am grateful for your continued support throughout the review process.